# On the Robustness of ML-Based Network Intrusion Detection Systems: An Adversarial and Distribution Shift Perspective

**Minxiao Wang** [1], **Ning Yang** [2,*,†], **Dulaj H. Gunasinghe** [1,†] **and Ning Weng** [1,*,†]

1   The Computer Engineering Program in the School of Electrical, Computer, and Biomedical Engineering, Southern Illinois University, Carbondale, IL 62901, USA; minxiao.wang@siu.edu (M.W.); dulaj.gunasinghe@siu.edu (D.H.G.)
2   The Information Technology Program in the School of Computing, Southern Illinois University, Carbondale, IL 62901, USA
*   Correspondence: nyang@siu.edu (N.Y.); nweng@siu.edu (N.W.)
†   These authors contributed equally to this work.

**Abstract:** Utilizing machine learning (ML)-based approaches for network intrusion detection systems (NIDSs) raises valid concerns due to the inherent susceptibility of current ML models to various threats. Of particular concern are two significant threats associated with ML: adversarial attacks and distribution shifts. Although there has been a growing emphasis on researching the robustness of ML, current studies primarily concentrate on addressing specific challenges individually. These studies tend to target a particular aspect of robustness and propose innovative techniques to enhance that specific aspect. However, as a capability to respond to unexpected situations, the robustness of ML should be comprehensively built and maintained in every stage. In this paper, we aim to link the varying efforts throughout the whole ML workflow to guide the design of ML-based NIDSs with systematic robustness. Toward this goal, we conduct a methodical evaluation of the progress made thus far in enhancing the robustness of the targeted NIDS application task. Specifically, we delve into the robustness aspects of ML-based NIDSs against adversarial attacks and distribution shift scenarios. For each perspective, we organize the literature in robustness-related challenges and technical solutions based on the ML workflow. For instance, we introduce some advanced potential solutions that can improve robustness, such as data augmentation, contrastive learning, and robustness certification. According to our survey, we identify and discuss the ML robustness research gaps and future direction in the field of NIDS. Finally, we highlight that building and patching robustness throughout the life cycle of an ML-based NIDS is critical.

**Keywords:** network intrusion detection systems; robustness; machine learning; adversarial attacks; distribution shifts

## 1. Introduction

Computer networks have revolutionized the way humans live, work, and communicate, and their continued success and advancement will undoubtedly shape the future of our interconnected world. With the development of computer networks, the attack surface has increased too. To protect networks from various security threats, many defense mechanisms against network attacks have been proposed, such as network intrusion detection systems (NIDSs). In recent decades, machine learning (ML) methods have been considered as a solution for solving intrusion detection problems.

ML has been widely applied in a broad range of industries and domains. For instance, ML applications in many domains, such as computer vision (CV) [1] and natural language processing (NLP) [2], have achieved significant success in the real world. At the same time, many network security tasks have also been built on the benefit of leveraging ML techniques. Recent NIDS advances [3,4] take advantage of deep learning (DL) to drive malicious network traffic detection and classifications. ML-based NIDSs can automatically

extract high-level features by learning from training datasets to achieve excellent detection performance and be more convenient than traditional signature-based NIDSs.

Despite the impressive performance of machine learning systems, their robustness remains elusive and constitutes a critical issue that impedes large-scale adoption [5]. Primarily for security tasks, such as NIDSs, robustness is the main concern for trustworthy real-world ML applications [6]. The considerable demand for robustness partially constrains the real-world implementation of ML-based NIDSs [7]. On one hand, research on the reliability and trustworthiness of ML-based NIDSs is still in the early stage [8,9]. On the other hand, numerous studies [7,10] highlight the concern that the vulnerability of applied ML will be part of the expanding attack surface. Furthermore, practical applications are crucial for validating theoretical advancements and gaining real-world insights [11]. In order to accelerate ML-based NIDS research with practical applications like CV and NLP, addressing the robustness of ML-based NIDSs should be a top priority.

In acknowledgment of the robustness requirement, an expanding collection of literature centers around the development and evaluation of robust ML systems [5] for not only NIDSs but also other fields. However, the increasing efforts at ML robustness are dispersed in various stages of the ML workflow and focus on different viewpoints [12]. Given that robustness in ML often entails multiple meanings depending on the context and use cases [13], a systematic survey on the state-of-the-art robustness studies for ML-based NIDSs is important.

In this paper, we aim to fill this gap by systematically assessing the advancements achieved so far on the robustness of the specific NIDS application task. Particularly, we investigate the robustness from the perspective of the capability of ML-based NIDSs in adversarial attacks and distribution shift scenarios. To gain insights into the robustness study of ML-based NIDSs, we analyze the similarities and differences between the robustness against adversarial attacks and distribution shifts through formulating and molding. Furthermore, we group the robustness studies by mapping them into different stages of the ML workflow to give a structured literature review. In addition, we highlight the research gap between NIDSs and other fields on the topic of robustness. Finally, we analyze the most prominent research trends within this field and compare the differences between NIDSs and other fields from the point of view of applying ML methods which will project into future research directions for robust ML-based NIDSs.

Our main contributions are as follows:

- We not only highlight the unique characteristics of ML-based NIDSs, and their relevance to robustness (Section 2.2) but also conduct an analysis of existing survey papers encompassing ML robustness and ML-based NIDSs (Section 2.3).
- We systematically summarize a taxonomy of existing ML-based NIDSs' robustness studies (Section 4.1). In our taxonomy, we arrange the robustness studies in six stages of the ML workflow. For each stage, we introduce research topics related to robustness challenges or robustness improvement methods for both adversarial attacks and distribution shifts aspects. In addition to the ML-based NIDS works, we also introduce some other fields' advanced ML studies and techniques.
- Based on our analysis, we summarize the main takeaways. We give some future research directions about the robustness of ML-based NIDS.

The rest of the paper is organized as follows. Section 2 introduces background related to studies among ML robustness, ML-based NIDSs, and existing robustness survey papers focusing on the NIDS task. The process of collecting valuable articles for our research topic is presented in Section 3. Section 4 presents our taxonomy of existing ML-based NIDS robustness studies, and more details about the two main robustness perspectives, adversarial attacks (Section 4.2) and distribution shifts (Section 4.3). Section 5 focuses on the inside robustness challenges and built-in methods for improving robustness. Section 6 focuses on the outside robustness challenges and patch-up methods for improving robustness. Section 7 gives the main takeaways and future directions. Section 8 concludes this work.

## 2. Background of ML Robustness, ML-Based NIDSs, and Existing Surveys

In this section, we give an overview of ML robustness background and identify the varying robustness-related terms among different scopes (Section 2.1). We highlight the unique characteristics of an ML-based NIDS and how they are related to its robustness (Section 2.2). We briefly summarize the existing survey papers related to ML robustness and ML-based NIDSs (Section 2.3). For readers' convenience, we summarize the notation table in Nomenclature.

### 2.1. The Concepts Related to ML Robustness

Robustness is a term that has become encompassed in a spectrum of interpretations and even overloaded [14]. For instance, robustness encompasses a wide range of aspects, including but not limited to raw task performance on test sets, the ability to sustain task performance on manipulated or modified inputs, generalization within and across domains, and resilience against adversarial attacks. Given the multifaceted robustness, we introduce the related concepts and present a concept tree to illustrate their relationship in Figure 1.

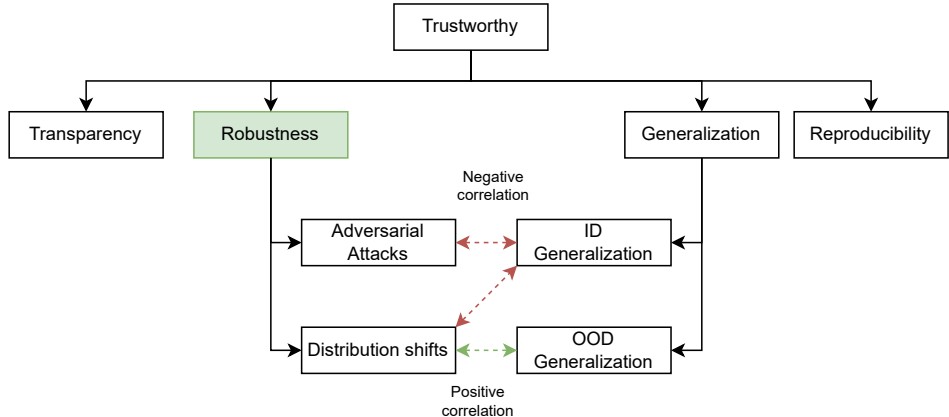

**Figure 1.** The concepts related to ML robustness. The red arrows refer to the negative correlation between the two concepts and the green arrow refers to the positive correlation.

**Trustworthy:** Trustworthy ML refers to ML models that are designed, deployed, and utilized in a manner that prioritizes ethical considerations, transparency (interpretation), accountability, fairness, and reliability (robustness). The robustness of ML corresponds to the reliability subfield of trustworthy ML.

**Generalization:** In the context of machine learning, generalization refers to a trained model's capacity to make accurate predictions on new, unseen data that were not part of its training set.

Depending on the data domain/distribution that the unseen data belong to, two cases of generalization are presented in the literature [15]. The first case is denoted as *in-domain (ID) generalization*, in which the unseen data are sampled from the same domain/distribution as that of the training dataset. For the second case, the model's capacity for correctly inferring unseen data that are sampled from a different domain/distribution is denoted as *out-of-domain (OOD) generalization*. Normally, the OOD generalization is basically the same as the robustness against distribution shifts.

**Distribution shifts:** distribution shifts refer to the phenomenon where the input data of ML models turn out to be different from the source distribution of the training data.

**Adversarial attacks:** adversarial attacks are a vulnerability of machine learning where deliberately crafted, small, imperceptible perturbations are added to input data, causing a trained model to misclassify or produce unintended outputs.

### 2.2. The Uniqueness of ML-Based NIDSs

ML has achieved numerous successes in recent years and maintained its influence across various fields, such as CV, NLP, and medicine. The shared element binding these

diverse domains is the abundant availability of data. Given the privacy concern, real-world network traffic data, which carry a wide range of sensitive information and valued business information, are not as readily available as in other areas. Meanwhile, the network traffic data's otherness, which is designed by humans and fully deformable (into tabular, images, or sequences), leads to a unique property of ML-based NIDSs—*varying data formats are adopted for ML-based NIDSs*.

*Tabular data:* The mainstream data format used for ML-based NIDSs is tabular data. Similarly, in the field of medical diagnosis, the main data format is also tabular data. However, we notice that there are still fundamental differences between tabular data in medical diagnosis and NIDS.

First, the subject of each line/sample of data is different. In medical diagnosis, the subject is different patients, which are all human beings. Despite differences in age, gender, and physical fitness, the relevant increase or decrease in one particular column of features has a similar meaning for diagnosis. However, the detection subject of NIDSs is network traffic flow, and different traffic flows may be dramatically different in most features. Additionally, the varying network environment will also affect the behaviors of traffic flows.

The second difference is varying features for the diagnosis of different diseases vs. a uniform feature set for detecting different attacks. The goal of medical diagnosis is normally to figure out which disease the patient has. Toward this goal, different test results are obtained, which directly affect feature column usage. The main benefit is that the features are strongly correlated with the diseases. However, NIDSs are required to use the same feature set for detection.

*Images:* Although network traffic can also be transformed into images in existing works, those byte images are different from the visual images in two aspects: First, the contents of byte images are not translation invariant. Unlike visual images, the contents in byte images have fixed locations in images. For example, the header information should always be at the top of the image. Second, there is no foreground or background in byte images. All of the contents in byte images are only parts of the raw bytes in the original network packets. For example, visual images are normally labeled based on the foreground, such as a picture of dogs. But the bytes images do not have the concept of foreground; all parts are combined into a whole.

Due to the varying data formats, the potential robustness challenges are different for NIDSs. On one hand, the ML-based NIDS methods using an image data format are vulnerable to spurious correlations. Those methods transform the raw network packet bytes into pixels of images; in this case, the payload of some malicious traffic flows, which are generated by the same attack tool, share a similar pattern. The pattern that is distinguished from normal traffic flows can be recognized as spurious correlations and hurts robustness. On the other hand, the ML-based NIDS methods using a tabular format are more sensitive to feature distribution shifts. Changing the deployment environment or temporal drift will cause significant performance degradation.

### 2.3. Existing Surveys of the Robustness of ML

The robustness of an ML model is critical for security applications such as NIDSs, as its failure can cause serious consequences on what is under protection. Therefore, unlike CV [1] or NLP [2] domains, a high-degree robustness is an essential requirement of ML-based NIDS for real-world deployment. Toward the long-term goal of deployment, this paper aims to bridge the gap in the systematical robustness study of ML-based NIDSs. In this section, we collect related existing survey papers based on their covering scopes among the topics that include *robustness*, *machine learning*, and *NIDSs*. We noticed that most existing surveys only focus on some aspects of robustness.

Adversarial attacks have received the most attention in the NIDS-related literature review papers. Apruzzese et al. [16] present a model for evaluating the realistic feasibility of adversarial attacks against ML-based NIDSs. Mbow et al. [17] provide a concise overview

and critical analysis of the recent advancements in the application of adversarial ML to NIDSs. They also discuss open questions that help define the future direction of this growing field. He et al. [18] investigate the gap between adversarial learning in the NIDS and CV domains. They achieved this by conducting a survey of the literature covering DL-based NIDS, adversarial attacks, and defensive techniques. The outcome is a thorough and encompassing portrayal of adversarial learning's role in the realm of DL-based NIDSs. Jmila et al. [19] conducted both a literature review and an empirical study. In addition to analyzing current challenges, they also evaluated the robustness of seven shallow ML classifiers and designed a Gaussian data augmentation defense technique. Beyond NIDSs fields, Sarker [20] presents various facets of AI-based modeling, including analytical, functional, interactive, textual, and visual AI. The goal is to grasp the essence of leveraging AI techniques effectively for automating cybersecurity, enabling intelligent decision-making, and ensuring robustness in security modeling. Adversarial learning is also examined within this context.

Regarding the distribution shift factor, concept drifts in ML-based streaming data models have received a thorough examination [21,22]. In many ML-based NIDS surveys [23,24], concept drift has been considered as a serious challenge; however, it has not been systemically reviewed in the context of NIDS. Besides concept drift, other types of distribution shifts, such as spurious correlations and covariate shifts, have not been comprehensive.

Hence, our objective is to address this void by conducting a comprehensive evaluation of the progress made thus far in enhancing the resilience of NIDS applications. Specifically, we undertake an exploration of robustness, focusing on the ability of machine learning-based NIDSs to withstand adversarial attacks and distribution shift scenarios.

## 3. Research Methodology

In this section, we illustrate the process of collecting valuable articles for our topic. This process included 3 steps: keywords for collecting literature, expanding the scope for a comprehensive coverage, and categorization and workflow mapping.

### 3.1. Keywords for Collecting Literature

We decided to focus our study on the robustness of machine learning-based network intrusion detection systems (NIDSs), specifically with a keen interest in adversarial attacks and distribution shifts, which is both timely and relevant to the field. However, in order to enhance the transparency and credibility of this research, it is imperative to introduce a dedicated section outlining the research methodology employed in the literature review process. First, we chose a group of keywords for searching articles. Three levels of keywords were chosen in this work: *core topic*, *scope and scenario*, and *technique*. For each level, the keywords are shown in Table 1.

**Table 1.** Three levels of keywords for the literature collection.

| Levels | Keywords |
| --- | --- |
| Core Topic | Robustness, adversarial, distribution shifts |
| Scope and Scenario | Machine learning, deep learning, neural networks, NIDSs |
| Technique | Poisoning attacks, evasion attacks, data augmentation, contrastive learning, adversarial training, fine-tuning, domain adaptation, robustness certification, cross-dataset evaluation, adversarial example |

### 3.2. Expanding the Scope for a Comprehensive Coverage

After the literature collection, we aptly acknowledged the scarcity of the literature explicitly addressing distribution shifts in ML-based NIDSs. To address this limitation and to provide a more holistic understanding of the subject matter, the authors wisely expanded

their scope to include applications beyond network intrusion detection. This includes areas such as computer vision (CV), natural language processing (NLP), and malware detection, which share similarities in terms of machine learning techniques and concepts.

### 3.3. Categorization and Workflow Mapping

We also found a lack of strong correlations between the final collected literature because most existing studies only focus on one particular technique or method to study, mitigate, or challenge one of the problems of ML/DL robustness. Considering that robustness is an inner capability of a trained ML/DL model, we try to split the different works based on their working stages. To better categorize and organize the extensive body of literature we found, we decided to map this literature into the workflow of machine learning. Detailing this process offers readers valuable insights into how the research was structured and enables them to follow the logical progression of ideas.

## 4. Taxonomy, Models, and Uniqueness of NIDS Robustness

In this section, we first present our taxonomy of NIDS robustness in Section 4.1. Then, we introduce more detailed knowledge of adversarial attacks (Section 4.2) and distribution shifts (Section 4.3). Finally, we give their definition and formulation in Section 4.4.

### 4.1. Taxonomy of NIDS Robustness Study

In this paper, we focus on investigating research that relates to the robustness of the ML-based NIDS model. Improving robustness necessitates coordinated efforts across multiple stages in the ML application life cycle, encompassing data sanitization, robust model development, anomaly monitoring, and risk auditing. Conversely, the breakdown of trust in any individual link or aspect can significantly compromise the overall trustworthiness of the entire system. Thus, a holistic approach to maintaining trust throughout all stages of the AI system's life cycle is essential to ensure its reliability and integrity [25].

Considering that ML model robustness is not a one-time achievement but an ongoing process that requires vigilance, updates, and evaluation, we organized our literature review sections (Sections 5 and 6) to follow the sequential stages of the ML workflow. As shown in Figure 2, we laid out the robustness-related research topics, which include both the challenges and solutions for adversarial attacks and distribution shifts, by the stages in which those studies mainly work. In the ML workflow, there are six main stages: (1) data collection and processing; (2) model structure design; (3) training and optimization; (4) fine-tuning (which is an optional stage); (5) evaluation; (6) application inference. From the point of view of model robustness, we considered obtaining the weights of models as a split point because once the training is finished, the robustness of the model is roughly settled down. Hence, we grouped the first three stages together for the reason that during those stages, robustness is built into the learning model. Furthermore, we grouped the remaining three stages together because the model robustness still can be patched up in those stages.

Investigating the ML-based NIDS model robustness, there are two major cases of models that we took into account in our work. *Case A*: an ML model that is well trained for a particular application network environment or scenario. *Case B*: an ML model that aims to learn general knowledge on intrusion detection. Due to different training and deploying purposes, the robustness of those two cases of models should meet different requirements.

### 4.2. Adversarial Attacks

Adversarial attacks aim to fool the ML model by perturbing the data [26]. Based on the different stages when the perturbed data are used, adversarial attacks can be classified into different types as follows.

**Taxonomy of Robustness study for learning-based NIDSs**

**Figure 2.** Taxonomy of the robustness study topics for ML-based NIDSs with topics grouped by their machine learning workflow stage. Two main robustness challenges, adversarial attacks and distribution shifts, encompass both the challenge and solution aspects.

- **Poisoning attacks:** In the training stage of ML workflow, poisoning attacks aim to perturb the training dataset by changing the inputs or shifting the labels so that they influence the trained model's future capability. If the attacker adds a trigger to the training data so that they can force the ML model to execute particular behaviors in the inference stage, those attacks are known as *backdoor attacks*.
- **Evasion attacks:** in the inference stage, evasion attacks refer to a type of attack that attempts to manipulate or exploit a machine learning model by perturbing input data in such a way that it confuses or misleads the model's predictions.

Based on the attacker's knowledge of target ML models, the adversarial attacks can be divided into three cases as follows:

- **White-box attacks:** The attackers know everything about the target ML models, such as the decision boundary. In this case, attackers can modify the inputs with the minimum perturbation but with a very high success rate [27].

- **Gray-box attacks:** the attackers only have part of the knowledge of target ML models and are able to access target models and observe their behaviors [28].
- **Black-box attacks:** the attackers do not have any information about the target ML models and cannot access the target models' responses.

Regarding ML-based NIDSs, adversarial attacks can be categorized into two types based on the level of input perturbation applied:

- **Feature-based attacks:** This type of adversarial attack against ML-based NIDSs focuses on perturbing the extracted features that represent a network traffic flow.
- **Traffic-based attacks:** Given the feature extraction component is included in NIDSs, it is impractical to directly modify the extracted features in real-world scenarios. Traffic-based attacks refer to those attack methods that focus on modifying the original network traffic [29].

### 4.3. Distribution Shifts

Distribution shifts will cause ML models to fail, such as being less accurate. Since the data are different from the source distribution, another term normally used to represent the robustness against distribution shifts is *out-of-distribution (OOD) generalization*. For varying data types, distribution shifts are normally classified into different subtypes [30] based on the causes.

*Tabular:* For many ML applications with tabular data, such as price prediction, there are three varieties of data distribution shifts [31]. Given inputs $X$ and their labels $Y$, the training data can be considered as a set of data samples from the distribution $P(X, Y)$. $P(X)$ denotes the probability density of the input, and $P(Y)$ denotes the probability density of the label. The label shift, covariate shift, and concept drift are each characterized as follows:

- A **label shift** arises when $P(Y)$ changes while $P(X|Y)$ remains constant.
- A **covariate shift** occurs when $P(X)$ changes while $P(Y|X)$ remains constant.
- A **concept drift** manifests when $P(Y|X)$ changes while $P(X)$ remains constant.

*Images and text:* For the real-world ML systems that work on image or text data, such as object detection, self-driving, and chat robots, even the foundation models pretrained on comprehensive large datasets are still likely unable to address the distribution shift issues [32]. Due to images and texts including richer background information than tabular data, the types of distribution shifts are more complicated. Two extra types are characterized as follows:

- **Spurious correlations** refer to statistical associations between features and labels that exhibit a predictive capability within the training distribution yet fail to constrain such predictive power within the test distribution [33].
- **Temporal (concept) drift and knowledge extrapolation** refers to language change and world knowledge change, which are unseen data far beyond the training distribution.

*Network traffic flow:* There are many factors that can cause distribution shifts in network traffic data, such as changing network environments, user behavior changing over time, and new advanced protocol versions. Additionally, given current ML-based NIDS methods work on varying types of data, including tabular [4], images [34], and sequences [35], the distribution shifts in network data have a *complex composition*. Although varying types of distribution shifts challenge the robustness of ML-based NIDSs, the studies related to the distribution shifts in ML-based NIDSs or network traffic analysis have not received enough attention. Existing works [36] only focus on one type of shifting cause, such as temporal drift.

### 4.4. ML Robustness Model

Robustness comprises both model-level and system-level aspects within the context of practical ML applications, such as ML-based NIDSs. At the model level, it involves reinforcing the resilience of the machine learning model itself. On a broader scale, system-

level robustness pertains to the entire application system, where the machine learning model assumes a pivotal role in delivering core functionalities. For instance, in systems like NIDS, the machine ML-based NIDS model forms an integral part of the overall application ecosystem. In this section, we focus on the robustness of the ML model, which is denoted as the capability of a trained model to withstand a multitude of dynamic challenges.

As per the definitions given in Section 4.2, in adversarial attacks, small perturbations ($\mathbf{r}$) are added to the input data. Hence, the robustness of the model is related to the smallest perturbation that needs to be given to the input data to change the output. Therefore, with adversarial attacks, the robustness can be defined as

$$\mathcal{R} = \mathbb{E}\left[\min_{\text{subject to } f(\mathbf{x}+\mathbf{r}) \neq y} \|\mathbf{r}\|\right]. \tag{1}$$

For the different types of distribution shifts presented in Section 4.3, we can find a mapping $T$. With the distribution shifts, a set of data might yield incorrect outputs while the remaining data points still yield the correct results. Hence, the robustness is related to the average shift of the inputs and inversely related to the average loss caused by the distribution shift. With distribution shifts, for a given (fixed) mapping $T$, the robustness can be defined as

$$\tilde{\mathcal{R}}(T) = \mathbb{E}[\|\mathbf{x} - T(\mathbf{x})\|] + \lambda \frac{1}{\mathbb{E}[l(f(T(\mathbf{x})), y)]}, \tag{2}$$

where $l(\cdot, \cdot)$ is the loss function, and $\lambda$ is a regularization parameter. Here, the second term is because if the mapping ($T$) gives a smaller loss, then the robustness is high, and vice versa. And the first term is because if the mapping ($T$) has to move the data point by a long distance to misclassify, then the robustness is high, and vice versa. Then, for distribution shifts, the overall robustness of the model can be defined as the minimum robustness of all mappings.

$$\mathcal{R} = \min_{T} \tilde{\mathcal{R}}(T) = \min_{T} \left\{ \mathbb{E}[\|\mathbf{x} - T(\mathbf{x})\|] + \lambda \frac{1}{\mathbb{E}[l(f(T(\mathbf{x})), y)]} \right\}. \tag{3}$$

In summary, both the margin defined in Equation (1) and the mapping defined in Equation (3) refer to the changes that happened to inputs. The defined formulations about robustness help to analyze the different robustness challenges and solutions in Sections 5 and 6.

## 5. Building in Robustness for Natural and Malicious Exploitation of Data Distribution Shift

As per Equations (1)–(3), training the model with the aim of maximizing the separation between data points and the decision boundary holds consistent benefits. This approach enhances the model's resilience against adversarial attacks and distribution shifts. To accomplish this objective, it becomes crucial to dedicate additional efforts toward the acquisition of well-balanced data, the augmentation of the original dataset, and subsequent training with these enriched samples.

### 5.1. Data Collection and Processing

Considering that ML-based NIDSs heavily rely on data, any inaccuracies during data collection and processing can inherently create vulnerabilities in terms of robustness. Hence, numerous studies aim to improve robustness during this stage too.

### 5.1.1. Adversarial Challenges and Response

The decision boundary of an ML model can be altered by attacking the training dataset. If the decision boundary lies in close proximity to the input data, then small perturbations to the inputs will lead to adverse outputs. As shown in Equation (1), if the input data can

be manipulated to provide adverse results by introducing very small perturbations, then the robustness of the model is very low.

**Poisoning attacks:** For the robustness against adversarial attacks, the most common challenge in the data collection and processing stage is the poison attack, which is a type of adversarial attack [37]. Poisoning attacks entail a form of attack wherein malicious entities manipulate the training data employed for constructing machine learning models. Due to NIDSs operating within the security domain, ML-based NIDS implementations inherently prioritize data privacy. As a result, considerations for data privacy are already integrated into their design. Consequently, unlike scenarios involving the creation of web-based open-world datasets or the utilization of online learning methods, the centralized offline learning approach of ML-based NIDSs remains resistant to the risks associated with poisoning attacks [38].

However, it is essential to acknowledge that the emergence of distributed technologies, such as federated learning (FL) and the Internet of things (IoT), introduces novel challenges related to data security and privacy. In these contexts, the decentralized nature of data aggregation and model training necessitates a careful consideration of potential data-related vulnerabilities. Nguyen et al. [39] introduce an innovative data poisoning attack, enabling adversaries to embed a backdoor within the consolidated detection model. This backdoor is designed for leading to inaccurately categorize malicious network traffic as benign. The adversary adeptly poisons the detection model over time, exclusively leveraging compromised IoT devices for injecting minimal quantities of malicious data into the training pipeline, while maintaining a covert presence.

To protect FL-based NIDSs from poisoning attacks, Zhang et al. [40] introduce an innovative and resilient FL-based NIDS named SecFedNIDS. This comprehensive approach comprises both model-level and data-level defensive mechanisms. At the model level, the authors present a strategic technique for selecting model parameters based on gradients. This method generates effective low-dimensional representations of locally uploaded model parameters. Additionally, they propose an online unsupervised approach for detecting poisoned models. In terms of data-level defense, poisoned data are detected by utilizing class path similarity, which is obtained through the layerwise relevance propagation method. Lai et al. propose DPA-FL [41], a dual-phase approach to defend against poisoning attacks. DPA-FL harnesses both relative comparison and absolute accuracy to swiftly mitigate the impact of poisoning attacks. The first phase, referred to as the relative phase (RP), identifies potential attackers by analyzing relative differences in weight between attackers and benign participants. The second phase, the absolute phase (AP), employs an accuracy assessment on a limited dataset. When the model's accuracy falls below a threshold, indicating susceptibility to an attack, AP can ascertain whether any attacker influences the global model.

*Discussion:* In summary, the research on poisoning attacks against ML-based NIDSs mostly focuses on FL and IoT scenarios. Compared with evasion attacks, poisoning attacks receive less attention. Obviously, launching poisoning attacks is more difficult than evasion attacks due to the absence of data access. We notice that the existing defense mechanisms have a common characteristic: protecting both data and models. To mitigate the impact of poisoning attacks, only protecting data may not be sufficient, because only altering a small portion of the training data will influence the NIDS model's behavior. Therefore, model training protection is required to prevent compromising the global ML model.

### 5.1.2. Distribution Shift Challenges and Response

As per Equations (2) and (3), if mappings can shift the data points by very small amounts leading to larger losses, then the model is said to have very little robustness.

For the robustness against distribution shifts, recent deep learning advances report that data augmentation can improve robustness/generalization under distribution shifts. Given the cost of data collection, data augmentation [1] is the simplest way to improve generalization using only currently available resources. However, we noticed that existing

data augmentation methods for NIDSs [42,43] mainly focus on solving the problem of imbalanced data. Therefore, in this subsection, we introduce some recent data augmentation studies which aim to help the robustness against distribution shifts in other fields.

**Data augmentation against distribution shifts:** A lot of research in the field of CV [44–46] and NLP [47,48] report that data augmentation can improve out-of-distribution robustness. However, due to the huge difference between network traffic and images or text, those methods may not be able to be directly applied to ML-based NIDS or other network security tasks. In this part, we introduce the general data augmentation methods for multiple data types or data augmentation network traffic data for improving robustness against distribution shifts.

In the field of DL-based encrypted traffic classification, Xie et al. [49] notice the challenge that although the classification performance achieved by existing deep learning models on encrypted traffic is impressive, a comprehensive study reveals a notable decline in their performance within varied and realistic network environments. To overcome this challenge, Rosetta was proposed to enhance the robustness of existing deep learning models for classifying TLS encrypted traffic. Rosetta focuses on perturbing the packet length sequences of flows, which is considered the main factor dramatically affected by varying TCP mechanisms and network environments. Rosetta consists of a TCP-aware traffic augmentation and a traffic invariant extractor. Three main TCP-aware traffic data augmentation methods are adopted, which are packet subsequence duplication augmentation, packet subsequence shift augmentation, and packet size variation augmentation.

Regarding the general data augmentation methods for multiple data types, Gao et al. [50] present a targeted augmentation method for OOD generalization. By theoretically analyzing the OOD risk for unaugmented models in a linear regression setting, they define four feature types for input, which are object feature, noise, robust feature, and spurious feature, based on whether those types are label-dependent and whether those types are domain/distribution-dependent. The target augmentation aims to randomize the spurious feature, which is dependent on the domain and independent of the label but preserves the robust feature, which is dependent on both the domain and the label. The targeted augmentation is evaluated on three real-world datasets across images of animals, biomedical images, and audio data.

*Discussion:* While addressing the issue of imbalanced data is indeed a crucial challenge within the realm of network security [51], it does not have a direct correlation with the topic of robustness. Data augmentation for imbalanced data focuses on improving the models' performance on those rare classes' data but not on improving OOD generalization, even though their methods for generating synthetic network data based on the original ones are essential for both types of network data augmentation.

Based on analyzing the characters of network traffic data, we believe that traffic data augmentation is challenging for the following reasons:

- *Mainstream data format—tabular data:* Most ML-based NIDSs use statistical features in tabular format for detection. But modifying the features' values is very risky, and it is hard to verify if the augmented samples are realistic or not.
- *Structured raw packets:* Network packets are designed for varying types of protocols and services. But within each individual type of traffic, the packet structure is clearly defined. That means unlike images, network data augmentation must only happen in the parts of raw packets that will not break the construction rules.
- *Flexible raw packets:* For network packets, not only the value of raw bytes can be modified but also the length of packets. The flexibility of each packet further exponentially affects the traffic flow they belong to. This flexibility makes it so hard to preserve the label-dependent features in the original data during augmentation.

*5.2. Optimization*

5.2.1. Adversarial Challenges and Response

**Contrastive learning with adversarial learning:** The performance of contrastive learning (CL) models depends critically on the design of positive and negative sampling strategies, and the robustness of the model will be greatly influenced by the difficulty of the suggested sample pairs. Self-supervised adversarial learning, as opposed to traditional CL, uses adversarial augmentation to make hard sample mining easier.

The data imbalance issue in network intrusion detection was addressed in [52] via adversarial data augmentation and self-supervised contrastive representation learning. In order to improve the representative learning progress in deep ML-based NIDS, the authors particularly presented a self-supervised adversarial learning method. This approach made use of an instancewise attack to produce a robust model by suppressing its adversarial susceptibility to perturbation samples. However, the performance discrepancy among different feature sets was obvious in the proposed research.

In [53], the authors suggest a novel method for improving the robustness of feature-space realizable adversarial examples-based ML-based Android malware detection. By identifying significant feature dependencies, the authors specifically provide a new understanding of domain constraints in the feature space. The authors utilize the optimum-path forest technique to identify these dependencies and use them to produce feature-space realizable adversarial examples during adversarial training in addition to taking statistical correlations into account. Domain knowledge is a valuable asset that may be used to improve the suggested strategy in the proposed research. However, a fundamental challenge is how domain experts' expertise might be used to set problem-space restrictions in the feature space.

5.2.2. Distribution Shift Challenges and Response

**Contrastive learning:** In order to glean information from the data themselves, contrastive learning creates pairings of positive and negative samples. Building a contrastive loss function is the fundamental concept behind contrastive learning. The model can compare similar and different data, draw on comparable samples, and draw out distinct samples.

In recent years, supervised CL has shown promising achievements in representation learning, natural language processing (NLP), and computer vision (CV) [54]. In [55], the authors put forth a supervised contrastive loss that performs better than cross-entropy loss and efficiently uses label information to group together point clusters that belong to the same class in the embedding space while isolating points from different sample classes.

In [56], the authors suggest a technique that contrasts network traffic in order to improve class-imbalanced learning in network intrusion detection. The dropout layer's randomness is used to produce various feature vectors, the model is inputted twice with the same flow, and supervised CL and cross-entropy are used to train the model. This method considerably enhances the learning of unbalanced network traffic and does not require the two phases of pretraining and fine-tuning, allowing for the wider exploration of harmful attacks concealed behind legitimate traffic. The acquired positive samples, however, are comparable when employing the dropout for data augmentation, which leads to feature suppression in the model. Since the model is unable to discriminate between sample similarity and class similarity, it is biased toward using a large number of typical traffic samples without taking into account their real variations in class.

In [57], the authors demonstrate how altering the feature space's characteristics might have an impact. For example, robust ML using feature-space models is extremely robust in content-based detection (which utilizes content rather than structural routes as features). Additionally, the authors demonstrate an improved feature-space model that uses conserved characteristics (which can be recognized automatically) and demonstrate that feature-space defense now succeeds where it previously failed. Additionally, they demonstrate the generalized robustness of feature-space techniques by demonstrating that the robust ML produced (after proper refinement using conserved features) is robust

against a variety of realizable attacks. However, one limitation of the proposed research is that the conserved features are limited only to the binary case.

## 6. Patching Up Robustness for Natural and Malicious Exploitation of Data Distribution Shift

As per Equations (1)–(3), the robustness against adversarial attacks and distribution shifts are all related to the distance between the data points and the decision boundary. For the fine-tuning, evaluation, and application inference stages, manipulating or measuring the distance can improve or evaluate ML models' robustness.

### *6.1. Fine-Tuning*

Pretrained models are typically refined through a process called fine-tuning. This involves leveraging the existing model parameters as a foundation and appending a task-specific layer trained from scratch on new data [58]. Fine-tuning is an optional stage in the ML workflow, which also will affect the assumption about robustness requirements. Considering both pretrained model and fine-tuned model, it should be noticed that they have different robustness requirements. The pretrained models, which are often trained on extensive and diverse datasets, are required to be robust to both distribution shifts caused by the environment and tasks and temporal concept drift. But the fine-tuned models should not be required to be robust to distribution shifts caused by different environments but should still be robust to adversarial attacks and concept drifts (i.e., an ML-based NIDS model is fine-tuned with a new dataset collected from the target network environment where this NIDS will work).

#### 6.1.1. Adversarial Challenges and Response

**Adversarial fine-tuning:** Wang et al. [59] propose Def-IDS, an ensemble defense mechanism against adversarial attacks by combining multiclass GAN-based data enhancement and multiclass adversarial retraining. Their results on the CSE-CIC-IDS2018 dataset demonstrate that Def-IDS has the capability to greatly enhance the resilience of deep ML-based network intrusion detectors against both known and unknown adversarial attacks.

Although retraining models with adversarial samples can improve robustness [59–61], how to generate the quality adversarial samples that can bring benefits is absent in those works. Fortunately, another research topic named robustness certification [62,63], which focuses on evaluating the robustness against adversarial perturbations, proposes methods to solve the problem. We introduce the details of robustness certification in Section 6.2.1.

#### 6.1.2. Distribution Shift Challenges and Response

**Domain adaptation:** Layeghy et al. [64] propose a cross-domain anomaly detection approach, DI-NIDS, which leverages domain adaptation techniques. Initially, a domain-adversarial neural network (DANN) is employed to extract domain-invariant representations of the data. This is achieved by incorporating a gradient reversal layer into the feature extraction network, which minimizes the dissimilarity between the representations of the source and target domains. During training, DI-NIDS uses both labeled source data and unlabeled target data to effectively train the DANN. Once the DANN is trained, DI-NIDS exploits its feature extraction branch to obtain domain-invariant features. These features possess the desired property of being insensitive to domain variations while retaining significant information. To accomplish the final objective of cross-domain anomaly detection, DI-NIDS applies a one-class support vector machine or one-class SVM (OSVM) [8] to the extracted features. This enables DI-NIDS to detect anomalies in a cross-domain setting effectively.

Qu et al. [65] introduce a novel network intrusion detection method utilizing domain confusion. The approach involves training a domain confusion network using generative adversarial networks (GANs). To address the issue of information loss during feature

transformation, a regularizer is incorporated. This helps achieve a domain-invariant feature representation of network traffic data that retains substantial information.

*Discussion:* Although many studies [66,67] report that fine-tuning can degrade a model's robustness, other NLP studies [68,69] show the opposite conclusion. In [68], the authors developed a novel fine-tuning method for NLP. In the fine-tuning step of the BERT model, supervised contrastive loss and cross-entropy loss were combined with weighting. A contrastive framework for self-supervised sentence representation transfer (ConSERT) was presented in [69], which used CL to fine-tune BERT unsupervised.

### 6.2. Evaluation

6.2.1. Adversarial Challenges and Response

**Randomized smoothing-based robustness certification:** Robustness certification studies aim to certify the robustness of an ML-based classifier against adversarial attacks/perturbations. One line of those works focuses on using the randomized smoothing technique to solve the problem [62]. The robustness certification has been successfully applied in CV [63,70] and NLP [60,61].

Recently, Wang et al. [71] proposed a robustness certification framework, named BARS, for DL-based traffic analysis, which can be considered as an upstream task of NIDSs. The design of BARS takes the unique attributes of the traffic analysis task into account, which are (1) the heterogeneous traffic data in the tabular format; (2) varied existing traffic analyzers; (3) serious adversarial application environments. BARS creates anisotropic optimized smoothing noise for input data, thereby achieving more stringent robustness guarantees through evaluating the traffic analyzers.

In detail, Wang et al. first extend the theory of robustness guarantee for CV tasks [62] to adapt it to network traffic analysis. Further, they design a distribution transformer to generate anisotropic smoothing noises for heterogeneous traffic feature inputs. Two optimization methods, noise shape optimizing and noise scale optimizing, are proposed to optimize the distribution transformer to achieve a better tightness of the robustness guarantee [63]. Finally, BARS uses the distribution transformer, which can automatically generate evaluating data to certify the robustness of the model in terms of robust region, dimensionwise radius vector, and the average dimension-heterogenous radius for the input data space. BARS can be applied for many different purposes, such as class-specific distribution transformer, noise data augmentation retraining, and robustness radius for the certification dataset.

*Discussion*: Randomized smoothing-based robustness certification methods utilize a large quantity of noised data samples to estimate the model's robustness against adversarial attacks. Therefore, it is similar to data augmentation for the reason both of them modify the original data samples. However, their differences are also significant. First, most data augmentation methods use a transform, which is fundamentally different from adding noise. Second, many data augmentation methods are experimentally proven widely effective, but how the randomized smoothing-based noise augmentation retraining precisely affects the risk of the robust smoothed classifier has not been comprehensively addressed. Recent work [72] reports that noise augmentation retraining's benefit can only be obtained by some distributions that have particular characters.

6.2.2. Distribution Shift Challenges and Response

**Cross-dataset evaluation:** Verkerken et al. [73] argue that anomaly-based NIDSs using ML see limited use in practical applications, even after years of research and development. They attributed this phenomenon to the inadequate generalization capabilities of the proposed models. Therefore, their article [73] uses a novel interdataset evaluation strategy to estimate the generalization of unsupervised ML-based NIDS models, such as principal component analysis (PCA), isolation forest (IF), autoencoder (AE), and one-class SVM. The interdataset evaluation strategy trains a model on the first dataset and evaluates it on the second dataset. The evaluation results show an average AUROC performance decrease

of 30.45% across CIC-IDS-2017 and CSE-CIC-IDS-2018 datasets. Their work remarks on the importance of improving the OOD generalization and claims their interdataset evaluation is a strong candidate for adoption in future research to estimate the generalization strength of newly developed models.

In [74], AI-Riyami et al. demonstrate that achieving high accuracy by using the recent performance evaluation strategy is easy for traditional machine learning and deep learning models but not practical for real-world applications. This observation motivated them to design a cross-dataset evaluation. Their subsequent work [75] proposed the correction and further empirically investigated cross-datasets evaluation for various machine learning methods in multiclass classification.

Apruzzese et al. [76] extend the cross-dataset evaluation from evaluating the robustness of trained ML-based NIDS models to evaluating the potential robustness of ML-based NIDS models. In short, mixing different datasets together not only happens at the testing stage but also at the training stage. First, a data-agnostic model is proposed for explaining the problem of cross-evaluation. Then, in order to overcome the challenges in cross-evaluation, a framework for cross-evaluating ML-NIDS, named XeNIDS, is designed. XeNIDS consists of four stages, which are *standardize*, *isolate*, *contextualize*, and *cross-evaluate*, so that it can achieve several benefits, such as removing network artifacts, mixing individual attack types labelwise, and enabling the development of collaborative ensembles of classifiers.

Layeghy et al. [77] claim that assessing the cross-domain performance of ML-based NIDSs represents a crucial yet inadequately explored gap in connecting the remarkable results achieved by the academic research community and the real-world implications of such research. Hence, they conduct a thorough investigation into the cross-domain performance of ML-based NIDSs. Their investigation involves comprehensive evaluations of eight supervised and unsupervised learning models using four recently published benchmark NIDS datasets. Meanwhile, they use the Shapley additive explanations (SHAP) values to show that feature importance order and the mean SHAP values are significantly different across datasets, which indicates different model behaviors.

*Discussion*: Training customized ML-based NIDSs is acceptable for certain crucial important network environments, even though sufficient data collection and annotation are highly costly. However, portability is also important for network protocols and services, as well as for ML-based NIDSs. From the point of view of ML research, training ML models to learn general knowledge, which can be reused in different scenarios is a long-term goal in every field. In this case, developing corresponding evaluation methods is essential to provide feedback information.

### 6.3. Application Inferences

Application inferences refer to the stage in which trained ML-based NIDS models are deployed in real-world application scenarios. During the application inference stage, the robustness of the deployed NIDS poses a severe threat from both malicious adversarial attacks and inevitable distribution shifts. For the adversarial attack aspect, numerous studies focus on adopting varying ML-based methods to generate attacks against NIDSs. On the other hand, many works propose solutions by introducing extra adversarial detectors to defend against adversarial attacks.

For the distribution-shift aspect, challenges that are raised by different causation are normally named separately; for example, the data shifting over time is named concept drift. Because the network is a dynamic environment and the network traffic is streaming data, concept drift received the most attention in NIDS distribution shift studies. Furthermore, the distribution shift also happens because of the major changes in the way NIDSs are used, such as different flow duration threshold settings in preprocessing. Meanwhile, studies on detecting and adapting the shifted inputs have been proposed to combat distribution shift issues.

### 6.3.1. Adversarial Challenges and Response

During the application inference stage, the robustness of ML-based NIDSs encounters a critical challenge known as evasion adversarial attacks. These attacks pose a severe threat to the security of the system, as they involve the deliberate manipulation of malicious traffic to evade detection and undermine the integrity of the target network. Recently, designing more realistic evasion attacks against NIDSs in the application inference stage received increasing attention. Realistic adversarial attacks are normally designed to work in traffic space based on practical assumptions about the real-world NIDSs' workflow settings. However, the study on how to protect NIDSs from adversarial attacks in the application inference stage has received limited attention.

**Evasion attack:** While finding inspiration from the field of computer vision (CV) [18], adversarial attacks targeted at NIDSs have been observed to exhibit differences in preprocessing and input space. Unlike adversarial attacks in CV, which directly perturbs the pixel values of images, adversarial attacks against NIDSs require perturbing data samples in feature space or traffic space.

However, feature-space adversarial attacks against NIDSs are impractical in realistic NIDS scenarios for several reasons. First, feature-space attacks require knowledge about the feature set employed by the target NIDS model. Second, the dependencies among adversarial features must be validated [78]; otherwise, the resulting adversarial features may be deemed invalid, as they might violate certain network domain facts, such as correspondence between ports and applications. Third, even the feature-space attacks still need to perform their effects by manipulating the raw traffic data in real-world environments.

For the traffic-space adversarial attacks, Sadeghzadeh et al. [79] proposed adversarial network traffic (ANT) that generated adversarial perturbation in three aspects of traffic space, packet payload length, packet number, and flow bursts. However, ANT required full knowledge of the target detection model and the feature set, and different perturbation operations were learned separately. Han et al. [80] proposed a two-step solution to practically generate traffic-space adversarial attacks against realistic scenarios. They first generated adversarial features with a GAN to let the malicious traffic mimic benign traffic in the feature space. Then, a particle swarm optimization (PSO) was adopted to project the feature perturbation back to the traffic space. Clearly, the two-step method incurred additional costs compared to the one-step approach, and it also required domain knowledge to guess the feature set for training the feature-space GAN.

Both Wu et al. [28] and Tan et al. [81] proposed reinforcement learning (RL)-based evasion attacks against NIDS models. However, RL-based methods require inspecting the feedback of target NIDS to train the RL models. Once their queries are blocked, they cannot finish training the adversarial RL models. An advantage of the attack in [81] is that their framework can perturb live network traffic, which makes their attack more practical in the real world. Similarly, Sharon et al. [29] proposed the TANTRA, which can end-to-end execute adversarial attacks by reshaping the original malicious traffic in the time domain. TANTRA trains an LSTM model to learn the temporal behavior of benign traffic within the interarrival time prediction task. The trained model is then used to generate new interarrival times for malicious traffic. TANTRA does not require any target model or feature set information, but it only perturbs the interarrival time. Another shortcoming is that the LSTM model has fixed outputs for specific inputs, which may result in the adversarial attacks having some pattern, which triggers other defense alarms.

**Adversarial example detection:** To defend against adversarial attacks in the NIDS models' application inference stage, adversarial example detection methods are proposed. Adversarial example detection aims to filter the adversarial examples before they are fed into the NIDS models.

Peng et al. design an adversarial sample detector in [82], which is based on a bidirectional generative adversarial network (BiGAN) [83]. First, the BiGAN is trained to learn the original clean data distribution (without adversarial perturbations) for reducing the adversarial noise and reconstructing the adversarial examples. Then, adversarial examples

are compared with the reconstructed samples to calculate the reconstruction error. Finally, the reconstruction error is combined with the error of the discriminator as the abnormal score, and when the score is larger, the input is more likely to be an adversarial example. Wang et al. proposed MANDA [84], a novel manifold and decision boundary-based adversarial example detection for ML-based NIDS. As the method name mentioned, two typical adversarial attack cases, the manifold case and decision boundary case, were targeted by MANDA. MANDA's detection philosophy is that an input is likely to be an adversarial example if it has an inconsistency between the manifold evaluation and the IDS detection or is very close to the decision boundary. Adversarial example detection is also included in Tiki-Taka [85], which is a comprehensive adversarial attack defense mechanism for NIDSs. Tiki-Taka assumes that the queries have inherent similarity; therefore, a deep similarity encoder (DSE) [86] is used to detect the received continuous queries, based on which the attackers learn to adjust the perturbations.

*Discussion*: Adversarial example detection is often considered the "last line of defense" because it operates after the model has been trained and deployed. Even if an ML model has undergone robust training, it can still be vulnerable to adversarial attacks. Adversarial example detection tries to catch such attacks at inference time. However, adversarial example detection for ML-based NIDS has not received enough attention.

Although adversarial example detection is important, it also has some shortcomings: (1) it requires extra adversarial example classifier, but NIDSs need to be efficient in monitoring network traffic; (2) existing works only evaluate feature-space adversarial attacks, which are based on some assumptions, and perturbed samples are distinguished from clean data [18], unlike the practical traffic-space attacks which have emerged recently.

### 6.3.2. Distribution Shift Challenges and Response

The possible reasons for distribution shifts (of different types) are: 1. Data representation (such as feature selection, processing configuration (artificial threshold)); 2. Data collection (human bias, incomplete collection); 3. Application scenario change; 4. Concept drift, $p(x \mid y)$ changes.

**Concept drift:** Concept drift refers to the phenomenon where the statistical properties of a target domain change over time in an unpredictable or arbitrary manner [87]. As real-time detection systems, ML-based NIDSs face the challenge of concept drift when monitoring network traffic streams. Different from adversarial attacks, concept drift normally is caused by some natural underlying changes in the higher-level environment. Therefore, to combat concept drift, ML-based NIDSs not only need concept drift detection but also concept drift adaptation. As a well-studied topic, many concept drift detection (window-based and performance-based methods) and adaptation (adaptive algorithms, incremental learning, and ensemble learning) methods have been proposed in other ML fields [22].

Recent NIDS studies on concept drift focus on designing comprehensive frameworks to improve the robustness against concept drift instead of working on detection or adaptation only.

Andresini et al. [88] propose a comprehensive ML-based NIDS to integrate both intrusion detection and concept drift detection together. They argue that both intrusion and concept drift detection should learn from the changes over time, but current ML-based NIDSs are built on the assumption of a stationary traffic data distribution. Their framework detects concept drift by the Page–Hinkley test (PHT) [89] and adopts incremental learning to update the training data and detection model.

Further, Andresini et al. propose INSOMNIA [90], which follows the underlying idea in [88], to combat concept drift and improve the model robustness at the same time. INSOMNIA leverages a DNN as its core classifier, and to mitigate the latency caused by model updates, it adopts an active learning approach, updating the model only with new points that yield maximum information gain. INSOMNIA also extends the goal of combating concept drift to also reduce the cost of labeling. INSOMNIA is designed as a

semi-supervised system, employing a nearest centroid neighbor classifier (NC) to estimate labels for the selected points.

Yang et al. [91] claim the retraining-based concept-drift defense methods are limited in practice for reasons such as "it is difficult to determine when the model should be retrained" and "Delayed retraining can leave the outdated model vulnerable to new attacks". They present CADE, which focuses on detecting and explaining each individual drifting sample. CADE adopts contrastive learning on the training dataset to learn a novel contrastive autoencoder-based concept drift detector. Furthermore, to explain the drifting samples in terms of feature importance, they design a new distance-based explanation method.

**Application scenario change (data distribution shifts):** In addition to concept drift, which mainly refers to temporal changes caused by dynamic environments, we believe that the more urgent area of study for ML-based NIDSs is the distribution shifts in the spatial view. Consider that most NIDS datasets are collected in particular environments but are expected to be used for training ML models that will be deployed in different environments.

Al-Riyami et al. [74] report cross-dataset evaluation results on the NSL-KDD [92] and gureKDD [93] datasets. Their results report a serious performance degradation of ML-based NIDSs when the testing data have a different distribution from the training data. They argue that NIDS research is conducted in such a way that training and testing the NIDS model in the same dataset is not practical for real-world application, because this type of evaluation performance cannot represent the quality of the models in the actual world.

Actually, the data distribution shifts caused by switching datasets are the same as the shift caused by an application scenario change. However, the application scenario change happens at different levels, for example, the high-level change from the NID in general Internet to the Internet of things (IoT), or the low-level change from a network environment belonging to universities to a network environment belonging to companies. Ideally, the long-term goal of ML-based NIDS studies is to build the capability of ML models to learn general and universal knowledge that can be easily reused for different scenarios.

*Discussion*: For ML-based NIDSs, improving robustness should consider both one-time learning (cross-domains) and lifelong learning (concept drift). One-time learning involves training a model on a varied dataset obtained from different domains or network environments. This is vital because real-world network traffic originates from diverse sources, each possessing distinct characteristics. If your NIDS is solely trained on a narrow dataset or domain, its performance could suffer when encountering unfamiliar and unanticipated data. Lifelong learning pertains to a model's capacity to consistently adapt and learn as the distribution of data evolves over time. Within the realm of NIDSs, alterations in network traffic patterns and attack methods (concept drift) can occur. A resilient NIDS must possess the ability to identify novel attack patterns that surface subsequent to the initial training period.

## 7. Research Summary and Future Directions

In this section, we expound upon the primary insights distilled from our comprehensive analysis and outline prospective avenues for advancing the resilience of machine learning-based NIDSs.

### 7.1. Main Takeaways

Based on our literature review and analysis in Sections 5 and 6, we summarize the main takeaways of this literature review in this section. The main challenges against the robustness of ML-based NIDS are summarized as follows:

- *Poisoning attacks* are not easy to launch against ML-based NIDSs. However, online learning and distributed learning systems (such as federal learning and IoT scenarios) are more vulnerable (Section 5.1.1).
- *Evasion attacks*, not only feature-based but also traffic-based, against ML-based NIDS have already received a lot of attention. However, how to use those attack methods to practically benefit robustness against adversarial is still unclear (Section 6.3.1).

- *Concept drift* caused by temporal change has been comprehensively studied for ML-based NIDSs. The main solution is the life-cycle adaptation method, specifically retraining the ML model after the drift happens (Section 6.3.2).
- *Distribution shifts* caused by a network environment change have received less attention than concept drift for ML-based NIDSs. However, a pretrained NIDS model that is generalized across different network environments will greatly benefit from being deployed in a particular environment (Section 6.3.2).

We summarize the main takeaways on the techniques related to ML robustness in Table 2. In this table, we compare different techniques in terms of impact on robustness, stages in the life cycle, degree of study in NIDSs, and degree of study in other fields, such as CV and NLP. We remark that those techniques impact robustness at both the ML model level and the system level. The system refers to a whole application system, in which the ML model plays the role of providing core functions (for instance, the NIDS system and the ML-based NIDS model).

**Table 2.** Summarized takeaways on the investigated techniques related to ML robustness.

| Techniques | Impacts on ML Model/System's Robustness | Stages in the Life Cycle | Degree of Study in NIDSs | Degree of Study in Other Fields |
|---|---|---|---|---|
| Poisoning attacks | Reduces model robustness | Data preparation | Moderate | Moderate |
| Evasion attacks | Unclear | Inference | Comprehensive | Comprehensive |
| Data augmentation | Improves model robustness | Data preparation | Limited | Comprehensive |
| Contrastive learning | Improves model robustness | Pretraining | Limited | Comprehensive |
| Adversarial training | Improves model robustness | Training/retraining | Moderate | Comprehensive |
| Fine-tuning | Based on the used data, could be beneficial or harmful | Retraining | Moderate | Comprehensive |
| Domain adaptation | Improves system robustness (against concept drifts) | Retraining | Moderate | Comprehensive |
| Robustness certification | Evaluates robustness (against adversarial attacks) | Evaluation | Limited | Moderate |
| Cross-dataset evaluation | Evaluates robustness (against distribution shifts) | Evaluation | Moderate | Moderate |
| Adversarial example detection | Improves system robustness (against adversarial attacks) | Inference | Limited | Comprehensive |

In addition, we noticed that contrastive learning and adversarial training were two methods that could be combined to train the ML model. Several research studies have been carried out on improving the robustness of NIDSs by training the model using both adverse and clean data via adversarial training. However, the current research on utilizing CL to improve the robustness of NIDSs is limited and needs further sophisticated investigations (Section 5.2).

### 7.2. Discussion on Future Directions

Based on Table 2, we further discuss the techniques that are limited in the current ML-based NIDS field but could bring potential opportunities for improving robustness. In this section, we focus on four techniques: contrastive learning, robustness certification, adversarial example detection, and data augmentation.

### 7.2.1. Contrastive Learning for NIDSs

In order to extend the typical supervised CL objective to self-supervised learning, which can learn with few labels, in the presence of class imbalance, and with better label-

independent initial feature information, novel research should be carried out focusing on automated feature extraction and data augmentation techniques for network traffic. The model can benefit from the pretraining and eventually learn a more generic representation of the network flow when the self-supervised learning conducts an effective initialization. Identifying meaningful conserved features in continuous feature spaces may be more challenging fundamentally. The extent to which modest differences in the list of recognized conserved characteristics matter is also an unresolved issue.

A better data-driven NIDS solution can be achieved by improving the representational ability of network flow data with a consistent and comprehensive behavior feature set. In addition, investigating a universal end-to-end method for more generic NIDS, which might significantly minimize the challenges of system implementation is also another future research direction. It might be anticipated that domain information can improve the precision of the search for feature dependencies. Hence, including domain knowledge to supplement data-driven methodologies in uncovering relevant feature dependencies is another intriguing path for future research.

### 7.2.2. Robustness Certification for NIDSs

Robustness certification presents substantial opportunities for the deployment of ML-based NIDS. This certification process theoretically establishes whether an ML-based NIDS model meets specific robustness criteria. Current robustness certification methods focus on quantifying DL models' robustness against adversarial attacks. Particularly, the robustness certification can estimate the robustness radius on testing data samples, which are generated by adding adjustable noise to original inputs. In addition to the mentioned randomized smoothing (Section 6.2.1), other methods, such as *α-CROWN*, *β-CROWN*, have not been explored for the ML-based NIDS field. Beyond quantifying and guaranteeing intramodel robustness, robustness certification can also serve as a continuous monitor to assist the extra adversarial example detector in filtering adversarial inputs before feeding them into ML models.

### 7.2.3. Adversarial Example Detection for NIDSs

Considering the trade-off between accuracy and robustness for a specific ML application scenario is well known [5], either accuracy or robustness decreasing is unbearable for a practical ML-based NIDS. Hence, an extra adversarial filter is essential, but some unique requirements must be taken into consideration.

A further research direction is how to design a real-time adversarial example detection. Otherwise, the extra component will become a new bottleneck hindering the efficiency of NIDSs. Another one is that adversarial example detection for NIDSs must be able to analyze both feature-based adversarial attacks and traffic-based adversarial attacks. Given the uniqueness of network traffic data (Section 2.2), the differences among varying data formats' adversarial attacks and attack detection methods should be considered when designing an adversarial example detection system.

### 7.2.4. Data Augmentation for NIDSs

The above-mentioned contrastive learning with adversarial examples, robustness certification, and adversarial example detection are all related to generating synthetic network data or adversarial network examples which can be denoted as data augmentation. Data augmentation is a widely employed technique across diverse ML tasks; however, network data augmentation is fundamentally hard because of the uniqueness of existing varying possible data formats for NIDSs. Therefore, we believe the future network data augmentation direction is to design comprehensive augmentation methods at the feature level, payload level, packet level, and traffic level.

## 8. Summary and Conclusions

In this survey, we collected, structured, and discussed literature related to the robustness of ML-based NIDSs from two perspectives: adversarial attack and distribution shifts. Based on the collected literature, we first systematically introduced the concept of ML robustness and its related concepts. Additionally, we discussed the uniqueness of ML-based NIDSs. Further, we designed a taxonomy to structure the adversarial attack- or distribution shift-related studies from both challenges and solutions viewpoints. In our taxonomy, we organized the reviewed papers according to which stage of the ML workflow the proposed method worked. For the related topics which have not received enough attention in the field of NIDS, we also supplemented the review with advanced works in other ML application fields such as CV and NLP. Finally, we presented the key insights derived from our analysis and outlined future research directions for investigating, measuring, and improving the robustness of ML-based NIDSs.

In conclusion, we argue that robustness should be considered at least as equally important as functional performance, such as accuracy. Given the essential aspect of ML robustness, building in and patching up robustness for ML-based NIDSs in their whole life cycle is necessary to guarantee their reliability in real-world deployment. We also would like to emphasize that robust pretrained NIDS models could be good starting points for building robust ML-based NIDSs. In the case of a pretrained NIDS model, the robustness should be more important than the accuracy for the reason that the NIDS task is vulnerable to concept drifts, such as zero-data attacks. We hold the perspective that the exploration of robustness is an ongoing endeavor. In the context of ML-based NIDSs, substantial endeavors remain essential to attain the threshold for practical real-world deployment.

**Author Contributions:** Methodology, validation, formal analysis, investigation, resources, writing—original draft preparation, visualization, M.W.; methodology, supervision, review and editing, project administration, funding acquisition, N.Y.; investigation, writing—original draft preparation, editing, D.H.G.; methodology, investigation, writing—review, supervision, N.W. All authors have read and agreed to the published version of the manuscript.

**Funding:** This research was funded by Ning Yang's startup funding and NSF award #2018919. In this work, Minxiao Wang is supported in part by Ning Yang's startup funding, and Dulaj Gunasinghe is supported by Ning Yang's NSF award #2018919.

**Conflicts of Interest:** The authors declare no conflict of interest.

## Nomenclature

| Acronyms | Meanings |
| --- | --- |
| AE | Autoencoder |
| ANT | Adversarial network traffic |
| CL | Contrastive learning |
| CNN | Convolutional neural network |
| CV | Computer vision |
| DANN | Domain-adversarial neural network |
| DDoS | Distributed denial of service |
| DL | Deep learning |
| IF | Isolation Forest |
| LSTM | Long short-term memory |
| MAC | Media access control |
| ML | Machine learning |
| NLP | Natural language processing |
| NIDSs | Network intrusion detection systems |
| OOD | Out-of-domain generalization |
| PCA | Principal component analysis |
| RNN | Recurrent neural network |

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
