# Peer review of "On the Robustness of ML-Based Network Intrusion Detection Systems: An Adversarial and Distribution Shift Perspective"

_computers, doi:10.3390/computers12100209_

Round 1
Reviewer 1 Report
In this paper, the authors reviewed the different issues related to the robustness of ML-based network intrusion detection. The overall presentation of the paper is well in shape. However, they should address the following issues:
Among the listed techniques, they should add self-supervised learning, semi-supervised learning, TL with domain-specific datasets, etc.
You may cite this paper which address the issues related to TL with domain-specific dataset: https://link.springer.com/article/10.1007/s11227-022-04830-8
Conclude the paper considering the above techniques.
Reviewer 2 Report
Dear authors,
I would like to thank you for your efforts composing this paper. Nevertheless, I have found areas for revisions and questions for clarification as follows:
- Figure 2 caption is overly long.
- After section 2, I highly recommend the authors to add a section on research methodology that sufficiently describes their approach towards categorizing and searching extant state-of-the-art literature. This will aid readers in grasping the techniques and methods you applied to achieve the goals of this paper.
- In Table 1, the two columns of “Degree of study” require further explanation. Hence, I recommend the authors to add footnotes defining each category. For instance, “well studied” should be described as a quantity of existing research studies (for example > 50). I also recommend the authors to cite studies for each category within the table.
I wish you all the best
Reviewer 3 Report
In this paper, Author aim to link the varying efforts throughout the whole ML workflow to guide the design of ML-based NIDSs with systematic robustness. My comments for the paper are:
• Can you explain the main challenges associated with using ML for network intrusion detection, especially in terms of robustness against adversarial attacks and distribution shifts?
• What is the current state of ML-based NIDSs in terms of their vulnerability to adversarial attacks and distribution shifts?
• You mentioned the need for comprehensive robustness throughout the ML workflow. Could you elaborate on what this means and why it's important?
• What are some common adversarial attack methods that ML-based NIDSs face, and how do they impact system security?
• How do distribution shifts in network traffic affect the performance and robustness of NIDSs, and why are they challenging to handle?
• could you summarize some of the key findings from your literature review regarding robustness challenges and proposed solutions?
• You mentioned advanced potential solutions like data augmentation and robustness certification. How do these techniques enhance the robustness of NIDSs?
• Author can read the following papers to increase the technical strength of the paper: An Intrusion Detection System Based on Normalized Mutual Information Antibodies Feature Selection and Adaptive Quantum Artificial Immune System
• Can you explain how data augmentation techniques can be applied to improve robustness in network intrusion detection?
Moderate editing of English language required
Reviewer 4 Report
In this paper, authors present related works on the robustness of ML-based NIDSs from two perspectives: adversarial attack and distribution shifts. A taxonomy is designed to structure the adversarial attack or distribution shifts related studies from those both challenges and solutions viewpoints. Finally, the key insights derived from analysis and future research directions for measuring, and improving the robustness of ML-based NIDSs are given. In general the paper has done a good job surveying the results and complete solution to the considered problem are presented through surveying and collecting various papers in the literature. The paper is also well written. The organization structure could be improved further but the paper is acceptable.
Some minor comments:
- Paragraphs with one sentences should be merged
- Organization of the paper can be improved further.
- Numbering bold descriptions, e.g. Concept Drift on page 16 would improve readability. Is it subheading?
- A notation table in the beginning would help.
English level is good.
Round 2
Reviewer 2 Report
Dear authors,
Thanks for your efforts improving the manuscript. I have no further concerns.
Kind regards
Reviewer 3 Report
accept